# Moisture Sensitivity Evaluation of the Asphalt Mortar-Aggregate Filler Interface Using Pull-Out Testing and 3-D Structural Imaging

Feng Xu [1], Xin Nie [1], Wenxia Gan [1,*], Hongzhi E [2,*], Peiyao Xu [3], Hongqiao Cao [1], Ruifang Gong [4] and Yuxiang Zhang [5]

1 Wuhan Institute of Technology, School of Civil Engineering and Architecture, Wuhan 430205, China; xufeng@wit.edu.cn (F.X.); deepmountain@126.com (X.N.); hideaheart@163.com (H.C.)
2 Highway Management Bureau of Luotian County, Luotian 438600, China
3 Anhui Road & Bridge Test Co., Ltd., Hefei 231283, China; hi_xupeiyao@163.com
4 Xinjiang Urban Construction (Group) Co., Ltd., Wulumuqi 830016, China; xjgongrf@163.com
5 Xinjiang Urban Construction Guoxin Engineering Co., Ltd., Wulumuqi 830016, China
* Correspondence: charlottegan@whu.edu.cn (W.G.); ehongzhi15261@163.com (H.E.)

**Abstract:** Moisture damage is one of the undesired distresses occurring in flexible asphalt pavements, mostly through water intrusion that weakens and ultimately degrades the asphalt mortar-aggregate interfacial bond. One method to mitigate this distress is using anti-stripping or anti-spalling filler materials that, however, require a systematic quantification of their interfacial bonding potential and moisture tolerance properties prior to wide-scale field use. With this background, this study was conducted to comparatively evaluate and quantitatively characterize the moisture sensitivity and water damage resistance of the interfacial bonding between the asphalt mortar and aggregate fillers. Using an in-house custom developed water-temperature coupling setup, numerous laboratory pull-out tests were carried out on the asphalt mortar with four different filler materials, namely limestone mineral powder, cement, slaked (hydrated) lime, and waste brake pad powder, respectively. In the study, the effects of moisture wet-curing conditions, temperature, and filler types were comparatively evaluated to quantify the water damage resistance of the asphalt mortar-aggregate filler interface. For interfacial microscopic characterization, the Image-Pro Plus software, 3-D digital imaging, and scanning electron microscope (SEM) were jointly used to measure the spalling rate and the surface micromorphology of the asphalt mortar and aggregate filler before and after water saturation, respectively. In general, the pull-out tensile force exhibited a decreasing response trend with more water damage and interfacial bonding decay as the moisture wet-curing temperature and time were increased. Overall, the results indicated superiority for slaked (hydrated) lime over the other filler materials with respect to enhancing and optimizing the asphalt mortar-aggregate interfacial bonding strength, moisture tolerance, and water damage resistance, respectively—with limestone mineral powder being the poorest performer.

**Keywords:** asphalt mortar; aggregate; filler; interface; water-temperature coupling; moisture sensitivity; water damage resistance; pull-out force; image processing; 3-D structural imaging

## 1. Introduction

Some of the premature damage and distresses that occur in traditional asphalt mixtures and flexible asphalt pavements are partially attributed to moisture intrusion [1]. These distresses typically manifest as loose aggregates, pulping, mesh cracking, peeling, pit grooves, stripping, cohesive degradation, loss of interfacial adhesion, debonding, etc. As reported in the literature [2–4], fillers are often used as anti-stripping or anti-spalling agents to enhance the moisture tolerance and water damage resistance of asphalt mixtures. The ASTM D 2425-17 [5] defines mineral filler as any finely grinded material meeting the

standard sieve sizes of 1.2, 0.6, 0.3 and 0.075 mm with a percentage pass rate of 100%, 97 to 100%, 95 to 100%, and 70 to 100%, respectively; with its total mass composition not exceeding 6% in the asphalt mixture. Whilst filler admixtures such as lime, cement, and amine anti-spalling agents have been successfully used to improve the adhesion of alkaline and acidic aggregates [5], slaked (hydrated) lime has, in particular, been utilized extensively to enhance the moisture tolerance and water damage resistance of asphalt mixtures and flexible asphalt pavements [6]. This wide usage in pavement construction is partly attributed to its (lime) cost competitiveness, readily availability, and eco-friendliness (i.e., relatively low production energy consumption and $CO_2$ emissions) compared to other filler admixtures such as cement.

The reviewed literature indicates that brake pads [7] contain steel slag [8], rubber [9], fiber [10], wear-resistant materials, etc., and that these constituent elements have a positive impact on the high temperature stability, moisture tolerance, and water damage resistance of asphalt mixtures. To enhance the asphalt-aggregate adhesion and improve the water damage resistance of asphalt mixtures, cement is also often used as a partial replacement of the limestone ore powder [11,12]. However, costs and environmental issues are some of the reported concerns to be cognizant with cement. In particular, the production of cement is associated with a substantial emission of $CO_2$ that detrimentally contributes to environmental pollution and global warming [13].

In general, two fundamental mechanisms and manifestations of water damage in asphalt mixtures have been reported, namely cohesive and adhesive failure [14]. Cohesive failure occurs within the asphalt itself due to the combined effects of oxidative aging, temperature fluctuations, and water intrusion, resulting in asphalt cracking between the aggregates [15,16]. Adhesive failure, on the other hand, refers to water entering the interface between the asphalt and aggregates, ultimately leading to a loss of adhesion between the asphalt and aggregates, with the asphalt stripping off from the aggregate surfaces [4,17,18]. At present, the commonly used methods to evaluate the moisture sensitivity and water damage resistance of asphalt and asphalt mortars/slurries alike include adhesion, surface energy, and rheological testing [15]. Using the asphalt adhesion bond strength test, Chaturabong et al. [14] successfully quantified the influence of moisture intrusion on the adhesion of the asphalt mortar with different filler materials. Hammet et al. [19] successfully used an atomic force microscopy to characterize the adhesive interface between asphalt slurry and different aggregate fillers. Francesco et al. [20], on the other hand, used a modified pneumatic adhesion tensile test device to assess and characterize the potential occurrence of three failure mechanisms in asphalt: namely cohesion, adhesion, and cohesion-adhesion mixed failure under both dry and wet (with moisture intrusion) conditions. The corresponding results indicated that cohesive and adhesive failure were each predominant under dry and wet conditions, respectively.

Various scanning and image processing tools are typically used to quantitatively analyze the interfacial debonding and damage characteristics arising from water intrusion. Commonly used among these software tools include Photoshop, Matlab, and Image-Pro Plus. The Photoshop [21] software is mainly used to edit and process pre-existing images with a focus towards special effects and artistic enhancements. However, this has a negative impact of diminishing the resolution and dimensional measurements of the image, which is not suitable for academic research applications. Whilst the Matlab [22] software is robust with respect to image morphological processing; it is not very accurate in terms of image quantitative analysis. Due its versatility and robustness, the Image-Pro Plus [23] software is routinely used in the medical and biological fields. Some of its superior characteristics include high image resolutions, more accurate dimensional measurements, numerical counting potential, classification statistics, and quantitative analysis ability [23]. Therefore, the Image-Pro Plus software was considered suitable for scanning/imaging the interface between asphalt and aggregates in this study. This was accomplished through novel modification of the image processing and analysis parameters in the Image-Pro Plus 6.0 software along with the integration of a 3-D precision structural lighting technique and Geomagic modeling module [24]. This innovative modification allowed the details of the

interfacial debonding to be accurately identified and captured more quickly/efficiently by photographing with a digital camera. Thereafter, the results of the interfacial debonding were quantitatively used to evaluate the adhesion properties, moisture sensitivity, and water damage propensity of the asphalt mortar-aggregate interface.

As documented in this paper, this laboratory study used an innovative, in-house, and custom developed pull-out test to quantitatively measure the bonding strength and water damage resistance of the asphalt mortar and aggregate interface with different fillers, namely limestone mineral powder, P·O 42.5 cement, slaked lime, and waste brake pad powder, respectively. The influence of different moisture conditions, water saturation time, temperatures, and filler materials on the interfacial bond strength were accordingly evaluated. As aforementioned, the Image-pro Plus recognition software, 3-D structural light digital scanning, 3-D model imaging, and scanning electron microscope (SEM) were collectively utilized to characterize the debonding, spalling (peeling off rate), and microscopic morphology of the asphalt mortar-aggregate filler interface. Overall, the primary goal of the study was to quantitatively evaluate and characterize the moisture sensitivity and water damage resistance of the interfacial bonding between the asphalt mortar and aggregate fillers. The second objective of the study was to comparatively assess and identify the best filler material that optimized the moisture tolerance, water damage resistance, and interfacial bond strength between the asphalt mortar and aggregates. In developing the subsequent work plans and test methods, it was envisioned that the study output will ultimately contribute to providing a reference datum for the characterization and use of bituminous (asphaltic) materials with aggregate fillers in extreme weather environments, particularly in rainy wet-cold regions.

Following this introduction and background section, the raw materials and test methods adapted for the study are discussed in the subsequent sections of the paper. Thereafter, the test results are presented, analyzed, and synthesized. The paper then concludes with a summary of key findings and recommendations.

## 2. Raw Materials and Test Methods

### 2.1. Test Materials

2.1.1. Asphalt (Asphalt-Binder, Binder, Bitumen)

A Class A #70 petroleum asphalt (namely Pen #70), sourced from Hubei Guochang, was used in the study. Its key physical properties and technical characteristics as evaluated using the Chinese standards JTG E20-2011 [25] and JTG F40-2004 [26] that are synonymous to the customary AASHTO (T 49, T 51, and T 53) and ASTM (D 113, 36, and D 5) standards, are summarized in Table 1. Note in this paper that the term "asphalt" interchangeably refers to "asphalt-binder", "binder", or "bitumen".

**Table 1.** Technical Indices for the Asphalt (Pen #70).

| Parametric Indicator | Penetration (25 °C, 0.1 mm) | Softening Point (°C) | Ductility @ 15 °C (cm) |
|---|---|---|---|
| Test results | 63.7 | 49.0 | >100 |
| Specification requirements [23] | 60 ~ 80 | ≥43 | ≥100 |

2.1.2. Filler Materials, Chemical Composition, and Packaging Density

The filler materials, namely limestone powder, P·O 42.5 cement, and slaked lime (i.e., hydrated or whitewashing lime) were sourced from the open commercial market. Brake pad powder was obtained from discarded brake pads that were crushed, ground (i.e., grinded), and sieved using appropriate shredders. The chemical composition of these four filler materials as quantitatively measured using the X-Ray Fluorescence (XRF) spectrometer [26–28] are listed in Table 2. Table 3, on the other hand, summarizes the packaging density of each filler material [29–31].

**Table 2.** Chemical composition of these four filler materials from XRF analysis.

| Composition | SiO$_2$ | CaO | MgO | Al$_2$O$_3$ | Fe$_2$O$_3$ | SiO$_3$ | K$_2$O | Na$_2$O | Ca(OH)$_2$ | BaO | Others |
|---|---|---|---|---|---|---|---|---|---|---|---|
| Proportion of the limestone mineral powder (%) | 5.8 | 36.8 | 1.4 | 1.8 | 0.4 | 0.1 | 0 | 0 | 0 | 0 | 53.7 |
| Proportion of the P·O 42.5 cement (%) | 30.1 | 41.1 | 1.5 | 12.5 | 3.4 | 2.9 | 1.0 | 0.6 | 0 | 0 | 6.7 |
| Proportion of slaked lime (%) | 0 | 0 | 0 | ≤2.0 | ≤0.05 | 0 | 0 | 0 | ≥96.0 | 0 | 0 |
| Proportion of brake pad powder (%) | 22.3 | 15.2 | 7 | 4 | 4.9 | 3.8 | 0 | 0 | 0 | 4.4 | 38.4 |

**Table 3.** Packing density of the fillers.

| Filler Material | Limestone Mineral Powder | P·O 42.5 Cement | Hydrated Lime | Brake Pad Powder |
|---|---|---|---|---|
| Apparent specific gravity | 2.639 | 3.072 | 2.325 | 2.235 |

In comparison to the other filler materials, Table 3 shows greater density for the P·O 42.5 cement, whilst brake pad powder had the least value [32]. In fact, P·O 42.5 cement's apparent specific gravity is about 20% higher than Table 3's overall average of 2.568 and about 13% lower the average in the case of brake pad power (i.e., lowest specific gravity) that comprise of relatively lightweight rubber.

2.1.3. Preparation of the Asphalt Mortar (Asphalt Slurry)

In this study, an asphalt-powder (filler) ratio of 1.0 to 1.0 was selected for the preparing the limestone mineral powder slurry. The other three fillers were used for the preparation of the asphalt slurry by equal volume replacement of the mineral powder. For preparing the asphalt mortar, the base asphalt (namely Pen #70) was heated to 140 °C. Thereafter, the dried fillers were slowly poured into the asphalt and mixed at a rotational speed of 2000 ± 200 rpm in accordance with the procedural recommendations by Bai et al. [33]. The asphalt was mixed with different fillers to obtain four different asphalt slurry (mortar) types with different fillers in volume proportions of 1.0 to 1.0. Lastly, the asphalt mortar was cured under different water saturation conditions. After moisture curing, one group of the asphalt mortar was placed in a 60 °C oven for one day and three days, respectively, to allow the filler to fully react with the asphalt—whilst another group was placed in a 60 °C water bath under water saturation conditions for a similar time duration, i.e., one day and three days, respectively, for comparison purposes to simultaneously assess the effects of water.

The Control group (i.e., the dry non-moisture conditioned speciemens), on the other hand, was setup and tested under normal ambient temperature conditions– that is after the asphalt mortar specimens were prepared and cooled down to room temperature (i.e., about 20 °C), testing was carried out immediately without any moisture wet-conditioning. For each filler material and moisture curing condition, a minimum of three asphalt mortar specimen replicates were prepared, molded, and fabricated. Like for the asphalt, note in this paper that the term "asphalt slurry" interchangeably refers to 'asphalt mortar".

2.2. Test Methods

2.2.1. The Pull-Out Test

As shown in Figure 1, a UTM loading frame (namely UTM-100) (IPCG, Hong Kong China) with custom designed and in-house made testing jigs, was used for conducting the pull-out test on the asphalt mortar with different fillers under the coupling effects of water and temperature. A 50 mm × 5 mm × 0.4 mm metal jig was custom fabricated in-house by these authors and placed on both sides of 50 mm × 50 mm × 10 mm limestone cubes—see Figure 1c. The asphalt mortar was then poured onto the 110 °C preheated

cubes, covered with another cube, and then pressed to a thickness of 0.4 mm to form the test specimen assembly shown in Figure 1d. Thereafter, the pull-out test, at a monotonic tensile pull-out rate of 10 mm/min, was conducted on the asphalt mortar specimens with different filler materials. The maximum pull-out force was captured and accordingly recorded to quantitatively characterize the adhesive bonding between the asphalt mortar and the aggregate (namely limestone cubes). Details of the test procedure is provided in the subsequent text.

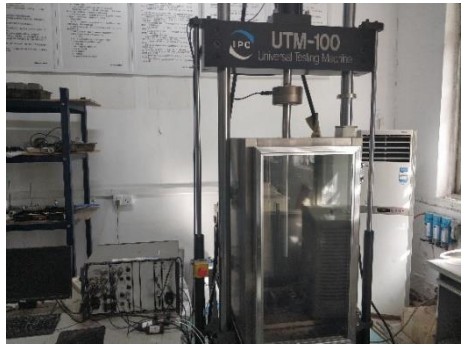

(**a**) UTM-100 testing machine

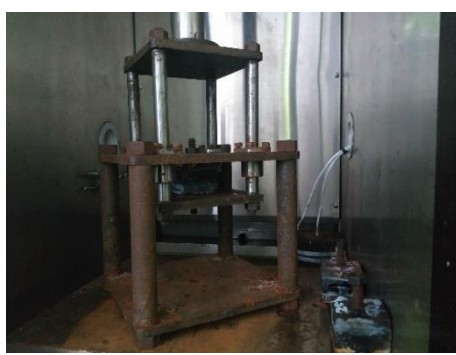

(**b**) Reaction support

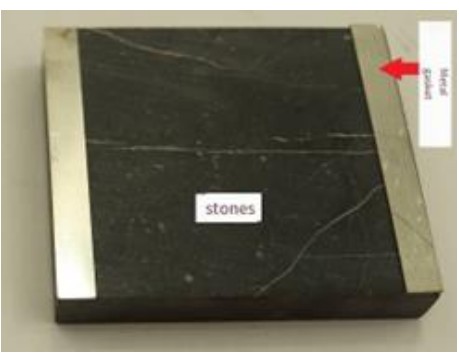

(**c**) Stone (limestone) and metal spacers

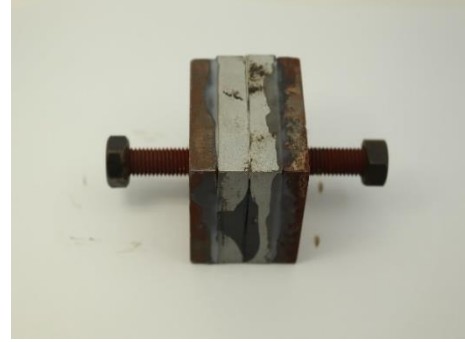

(**d**) Test specimen assembly

**Figure 1.** The pull-out loading configuration and test sample setup.

As custom-devised by these authors, the test procedural sequence involved firstly cutting the limestone blocks into small limestone cubes of 50 mm × 50 mm × 10 mm and sanded to the same roughness/flatness using uniformly meshed sandpapers. Thereafter, a 50 mm × 5 mm × 0.4 mm thin aluminum sheet was placed on both sides of the limestone cubes (i.e., stone cubes) as exemplified in Figure 2c, with the resultant asphalt mortar-aggregate interface assumed to correspond to an asphalt film thickness of 0.4 mm. The 135 °C heated asphalt mortar was then poured into a 110 °C heated cube, covered with another 110 °C heated cube on top, and then pressing the pair to a thickness of 0.4 mm. After about 2 h of room temperature cooling, the oozing asphalt mortar on both sides of the cubes was scraped off to form a "sandwich biscuit" shaped test specimen assembly shown in Figure 2d. Thereafter, the test specimen assembly was fixed onto the UTM loading frame using some in-house custom-made support fixtures and then preconditioned at least for 4 h prior to testing. The preconditioning temperatures were 0, 5, 10, 15, and 20 °C, respectively. That is five different sets of the test specimen assemblies were fabricated, and each set was separately tested at one of these five temperatures. The test was automatically set to terminate within 75% of the UTM axial sensor range, by which time fracture failure of the interface had occurred, with the peak tensile load dropping to zero. Once the test was completed, the damaged/fractured interface was photographed using a digital camera and 3-D structural light image scanning. Thereafter, the exfoliated area of the asphalt mortar

on the fractured interface was estimated and quantified using the Image-Pro Plus 6.0 and Geomagic Wrap software (version 2021) [33,34].

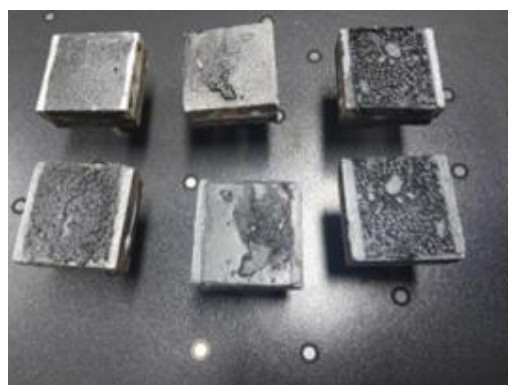
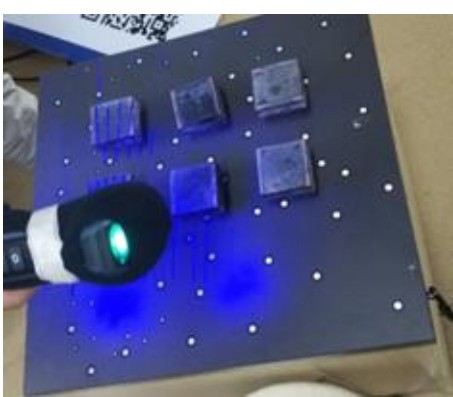

(**a**) Placement of specimens          (**b**) Scanning of the specimens

**Figure 2.** Structured light scanning of the asphalt mortar specimens.

To evaluate the bonding strength of the asphalt slurry with different fillers and adequately capture the rate of the pull-out force more directly, different water-temperature coupling effects were investigated on the asphalt mortar specimens. In this study, the magnitude of the monotonic tensile pulling force and reduction rate (as illustrated in Equation (1)) were used as the quantitative indicators of the moisture sensitivity and water damage resistance of the asphalt mortar-aggregate interface [35,36].

$$\eta_L = \frac{F_w - F_i}{F_w} \times 100\% \tag{1}$$

In Equation (1), $\eta_L$ is the reduction rate of the monotonic tensile pulling force; $F_w$ is the pulling force at room temperature without any water saturation; and $F_i$ is the pulling force after curing with water at specified times and temperature conditions [36].

2.2.2. The Peeling Rate Test: Structured Light 3-D Scanning

Structured light 3-D scanning technology has been widely used in the fields of national defense, aerospace, industrial design, mold manufacturing, precision engraving, rehabilitation, and medical treatment, but it is rarely used in asphalt mortar applications [36]. In this study, structured light 3-D scanning was exploratorily used to study the spalling rate of the asphalt mortar with different fillers. The technical objective was to quantitatively evaluate the anti-spalling performance of the asphalt mortar. The methodology incorporated the following procedural steps:

Step 1—image scanning. After the pull-out test was completed, a hand-held structured light 3-D scanning device (Artec 3D, Senningerberg, Luxembourg) with a precision of 0.05 mm was used to conduct rapid scans of the structural plane of the tested asphalt mortar specimens. For each specimen, the test involved repeated scans at multiple angles, as shown in Figure 2.

Step 2—image processing. The 3-D data of the structured light 3-D scans were imported into a Geomagic Wrap software [34] as exemplified in Figure 3. Thereafter, the 3-D model was initially modified, including recalibration of the initial position and deletion of some excessive redundant parts, to ensure reliability/accuracy of the quantitative analysis and final image results.

Step 3—selection and calculation of the exfoliation section. Multilayer horizontal exfoliation sections parallel to the interface with 0.2, 0.4, and 0.6 mm distances were established, and the area above each exfoliation section was obtained. The exfoliation rate of the asphalt mortar (slurry) on the exfoliation surface image of the specimens was calculated using Equation (2).

$$\text{Spalling rate of the asphalt mortar} = \frac{\text{The number of pixels of exfoliated asphalt mortar}}{\text{The total number of pixels on the complete stone surface}} \times 100\% \qquad (2)$$

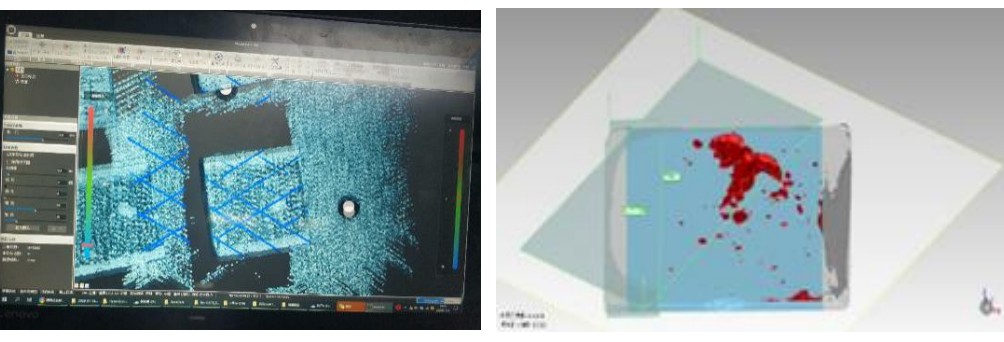

(**a**) Preliminary treatment specimen model　　　　　　　(**b**) Cut section

**Figure 3.** Geomagic Wrap software image processing.

### 2.2.3. The Peeling Rate Test: Image-Pro Plus Software Treatment

For better evaluation of the asphalt mortar's water damage-resistance characteristics, a digital camera was used for taking pictures as well as processing the peeling off images of the tested asphalt mortar specimens. A correlation analysis of the structured light 3-D scanning results was conducted to obtain a rapid and more accurate methodology for evaluating/quantifying the peeling resistance of the asphalt mortar. As discussed below, the methodological approach encompassed two key phases, namely: (a) image acquisition, and (b) image processing and calculations.

In the first phase of image acquisition, the fractured interfaces of the pull-out tested specimens were photographed with a digital camera and the corresponding images were processed using the Image-Pro Plus software [23] to determine and quantify the peeling rate of the asphalt mortar with different fillers. To overcome the influence of uneven illumination and shadows caused by the digital camera, a special photo box was used for image acquisition. After the pull-out test was completed, the tested specimens were placed in a special photo box shown in Figure 4. The digital camera's photographical shooting parameters were fixed at ISO-100, with exposure time and aperture values of 1.0 s and 29, respectively.

After image acquisition, the second phase of image processing, computational modeling, and analytical calculations was implemented. That is, after the spalling images of the fracture surfaces of the pull-out tested specimens were collected, the photographical parameters such as the spalling area of the asphalt in the image were intricately extracted to quantitatively evaluate the adhesion between the asphalt mortar and aggregates—which is discussed in the subsequent text. The complete experimentation process starting from specimen preparation through testing to modeling analysis, imaging, and analytical calculations is graphically and visually summarized in the flow chart in Figure 5.

As graphically illustrated in Figure 5, some of the final steps of determining the spalling rate involved image processing, corrections, and measurement of the target objects. The specific procedural steps to obtaining the spalling rate of the asphalt mortar and aggregates interface using the Image-Pro Plus software [24] are as follows:

Step 1—image preprocessing. The color saturation of the shot image is generally low. So, the gray units of the system were converted into optical density units through Photoshop format transformation to reduce the systematic errors associated with the image. Thereafter, the color ratio of the image was adjusted using the Image-Pro Plus software [24]. The HIS color format was used for color separation selection, where H is the hue, S is the saturation, and I is the intensity. When selecting the AOI (Automated Optical Inspection), first keep H and S as the maximum range and then select the red component from I to reduce the systematic errors associated with the image. Refer to Figure 4 for an example of a processed image.

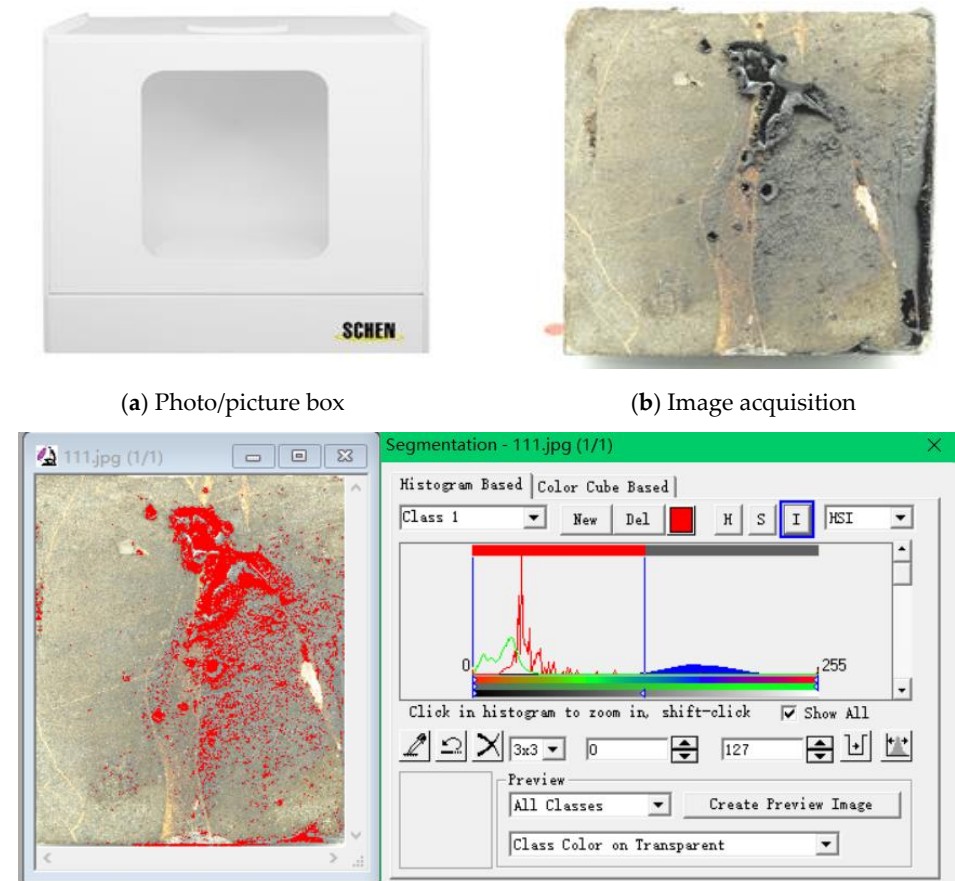

(**a**) Photo/picture box                                    (**b**) Image acquisition

(**c**) Image-Pro Plus 6.0 software processing

**Figure 4.** Image acquisition and processing with the Image-Pro Plus software.

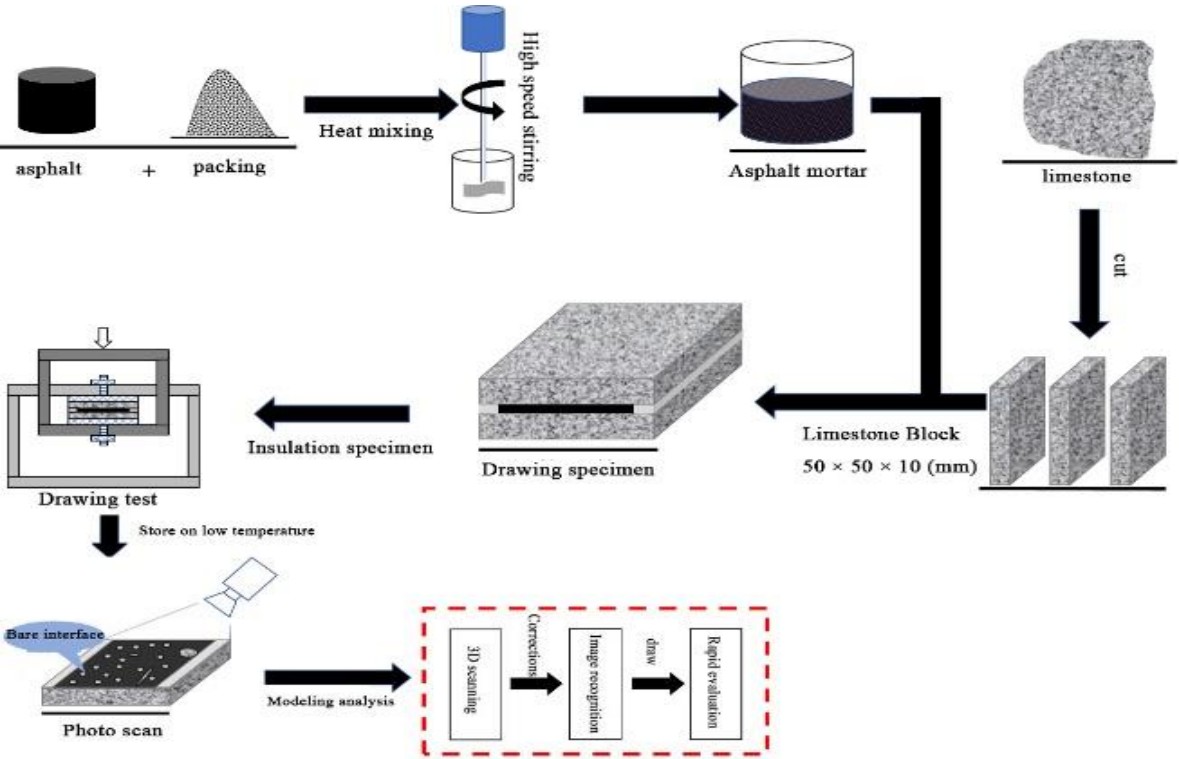

**Figure 5.** Flow chart for the complete process (i.e., specimen preparing, testing, imaging).

Step 2—corrections and measurements of the target object. After selecting the target image in Step 1, the selected target area should be modified to eliminate any wrongly selected parts to improve the accuracy of target selection. Thereafter, the pixel statistics function of the Image-Pro Plus software [24] was used to count the number of pixels of the target image as well as the total number of pixels of the complete aggregate surface.

Step 3—calculation of the spalling rate of the asphalt mortar on the aggregate surface. Similarly, the spalling rate of the asphalt mortar on the image of the spalling surface of the pull-out tested specimens was calculated using Equation (2), which was used as the basis for quantitatively evaluating the anti-spalling performance of the asphalt mortar.

### 2.2.4. The Peeling Rate Test: Parametric Modifications and Pearson Statistical Correlations

Pearson statistic correlation analysis [37–39] was performed between the multi-section peeling data obtained using structured light 3-D scanning [40] and the peeling rate obtained using the Image-Pro Plus software [23] to obtain more accurate peeling results. The parametric correction and correlation process are illustrated in Figure 6.

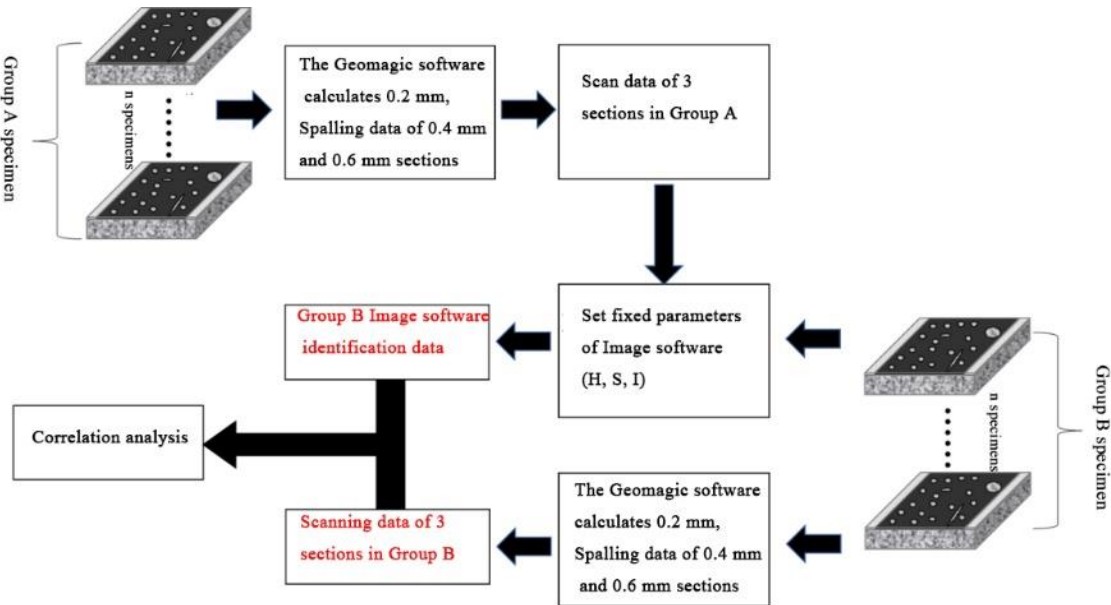

**Figure 6.** Parametric modifications and correlation process.

### 2.2.5. Scanning Electron Microscope (SEM) Imaging Analysis

In this study, the morphological macrotexture and microstructure of the interface between the asphalt and aggregate surfaces was analyzed using the Scanning Electron Microscope (SEM) (CIQTEK, Hefei, China) as a function of different filler materials [30]. SEM imaging of the pull-out tested specimens was conducted before and after water saturation to microscopically characterize the moisture sensitivity, water damage resistance, and anti-spalling potential of the asphalt mortar in terms of the surface micromorphology.

## 3. Test Results and Discussions

### 3.1. Pull-Out Test Results and Analysis

### 3.1.1. Temperature and Filler Effects on the Pull-Out Force and Displacement

To assess and quantify the effects of temperature on the pull-out force and interfacial bond strength, the test specimens with different fillers were temperature conditioned for more than 4 h after being saturated in water at 60 °C for 1 day. The five 4 h conditioning and test temperatures were 0, 5, 10, 15, and 20 °C [41], respectively. That is for each of these five temperatures, the asphalt mortar specimens were preconditioned for at least 4 hrs prior to pull-out testing. Due to its viscoelastic nature, the asphalt's adhesion and bonding characteristics are traditionally considered more critical at low temperatures and

hence, the selection of low-test temperatures of 20 °C and below in this study. The lower test temperature of 0 °C was specially included in the matrix to exploratory aid with simultaneously assessing the effects on the bonding strength and response behavior of the 60 °C-1 day moisture conditioned specimens under freezing temperature conditions to simulate the extreme winter seasons. On the other hand, the 20 °C represented the ambient temperature and served as the reference datum, i.e., the Control. At high temperatures, asphalt (due to its viscoelasticity) tends to get soft with improved adhesivity and bonding propensity. Therefore, the authors found it unwarranted to include higher test temperatures exceeding 20 °C in this study—but will be explored in future follow-up studies [41]. The generated force-displacement response curves, for different temperatures, are plotted in Figure 7.

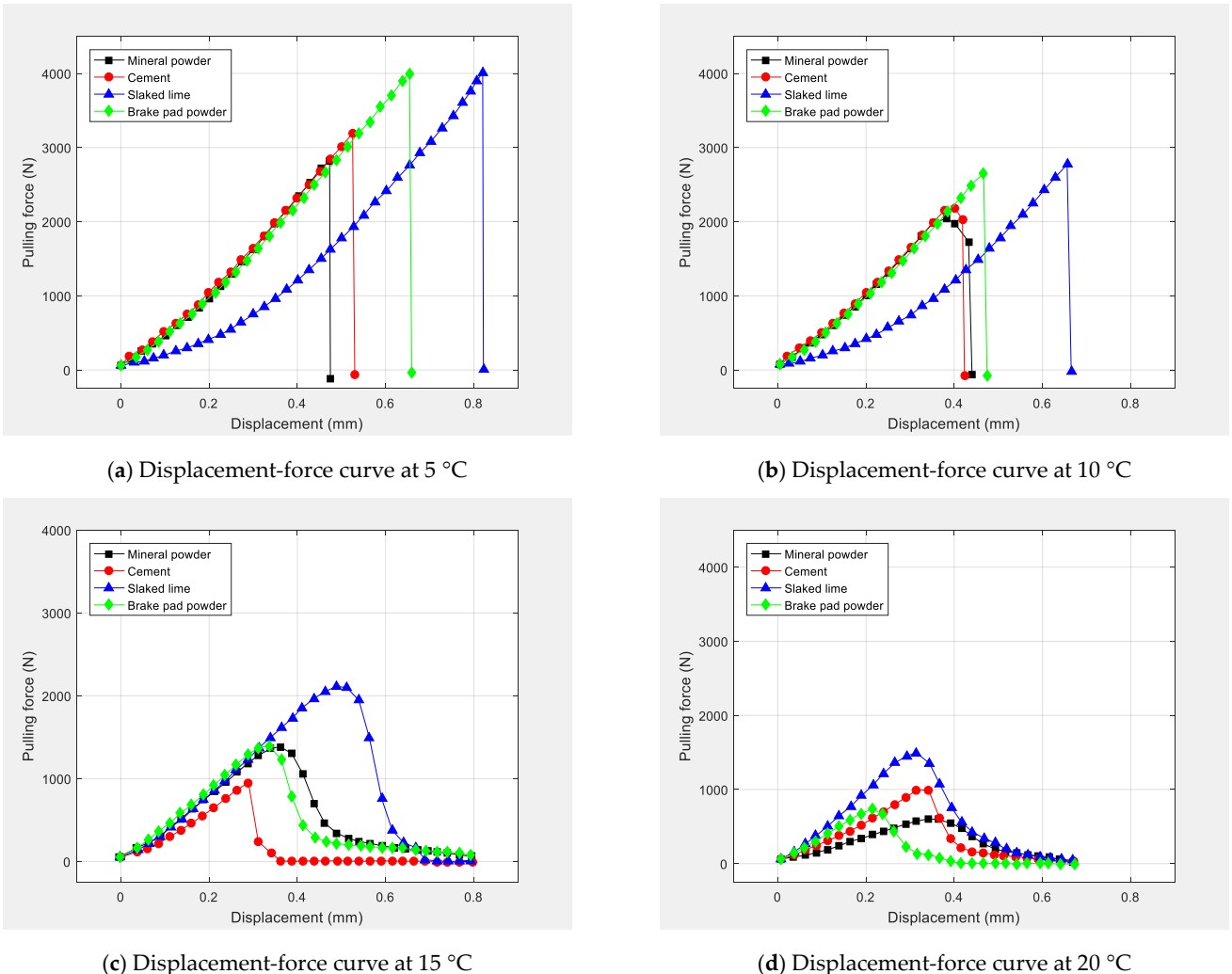

(**a**) Displacement-force curve at 5 °C

(**b**) Displacement-force curve at 10 °C

(**c**) Displacement-force curve at 15 °C

(**d**) Displacement-force curve at 20 °C

**Figure 7.** Force-displacement response curves of the asphalt mortar at different test temperatures.

It can be observed in Figure 7 that the tensile pulling force is inversely proportional to the test temperature and decreased with an increase in the test temperature. However, whilst the interfacial pull-out force decreased rapidly (i.e., instantaneous) within the first test temperature range of 5 to 15 °C, the decline was much slower for the 15 to 20 °C temperature range. Likewise, the viscosity (stiffness) of the asphalt mortar (due to the viscoelastic nature of the asphalt) was also observed to be inversely proportional to the test temperature [15,42,43]. That is the higher the temperature, the lower the stiffness and the lower the peak tensile failure load. During interfacial pull-out testing, instantaneous

fracturing of the asphalt mortar is thus more likely to occur at lower temperatures, making the interface failure to change from adhesive to cohesive as the temperature increases.

Furthermore, it was observed that under the low temperature conditions of 5 and 10 °C, the asphalt within the asphalt-mortar was more in the elastic phase with a relatively high stiffness, leading to brittle adhesive-fracture failure accompanied by an instantaneous load drop to zero—see Figure 7a,b. As evident in Figure 7c,d, this was not the case for 15 and 20 °C as the asphalt was shifting towards the softer viscous phase, with relatively low stiffness and peak loads that gradually reduced to zero—suggesting more of a ductile (i.e., cohesive-fracture) failure than brittle adhesive-fracture failure. When the test temperature was 10 °C, however, there was a transitional indication of both adhesive and cohesive failure of the asphalt mortar in the force-displacement response curves for mineral powder and cement—see Figure 7b. As photographically illustrated in Figure 8a, this suggests that brittle fracture failure predominantly occurs at low test temperatures such as 5 °C. At around ambient temperatures (i.e., 15 and 20 °C), the asphalt mortar interaction exhibits a cohesive mode of failure, with the resulting fracture interfaces exemplified in Figure 8c,d, respectively. By contrast, however, the fracture interfaces in Figure 8b visually indicates the simultaneous occurrence of both adhesive and cohesive failure occurred at 10 °C, which is synonymous to the limestone mineral powder and cement's response curves in Figure 7b. By and large, these interfacial photographs of the fracture interfaces in Figure 8 somewhat corroborates the graphical response trends of the asphalt mortar observed in Figure 7.

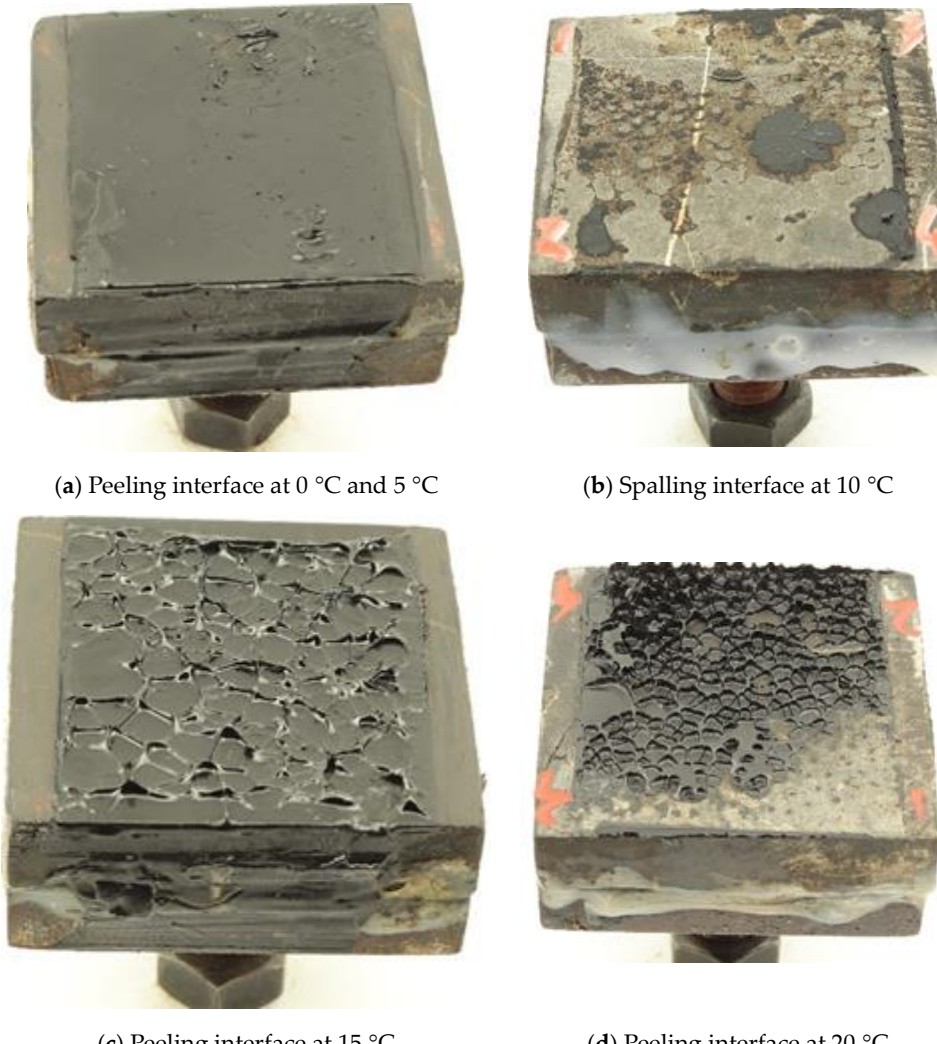

(**a**) Peeling interface at 0 °C and 5 °C    (**b**) Spalling interface at 10 °C

(**c**) Peeling interface at 15 °C    (**d**) Peeling interface at 20 °C

**Figure 8.** Failure pictures of the fracture interfaces of the asphalt mortar at different temperatures.

Looking at the magnitudes of the pulling tensile force and displacement in Figure 7, it is evident that slaked lime generally exhibited performance superiority over the other filler materials at all the test temperatures that were evaluated. At the lower temperatures (i.e., 5 and 10 °C), the limestone mineral powder performed the poorest with the least pulling tensile force and failure displacement, respectively. At 15 and 20 °C, the poorest performers were cement and brake pad powder, respectively.

### 3.1.2. Water-Temperature Coupling Effects on the Interfacial Bonding Performance

For assessing the interactive effects of water and temperature coupling, the moisture curing conditions were dry (no $H_2O$) and full water saturation for 1 day and 3 days at 0 and 60 °C, respectively. Thereafter, the water-temperature coupling tests were conducted at a test temperature of 10 °C and monotonic pulling (tensile) loading rate of 0.1 mm/min. The corresponding results of these tests are graphed in Figures 9 and 10.

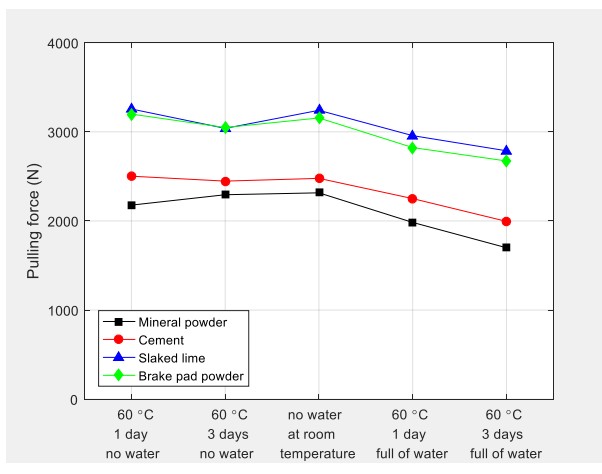

(**a**) Force vs. moisture curing at 0 °C

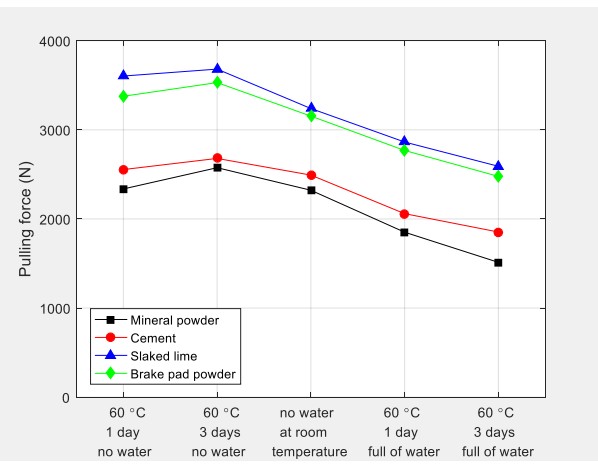

(**b**) Force vs. moisture curing at 60 °C

**Figure 9.** Pulling force results as a function of moisture ($H_2O$ saturation) curing time at 0 °C and 60 °C.

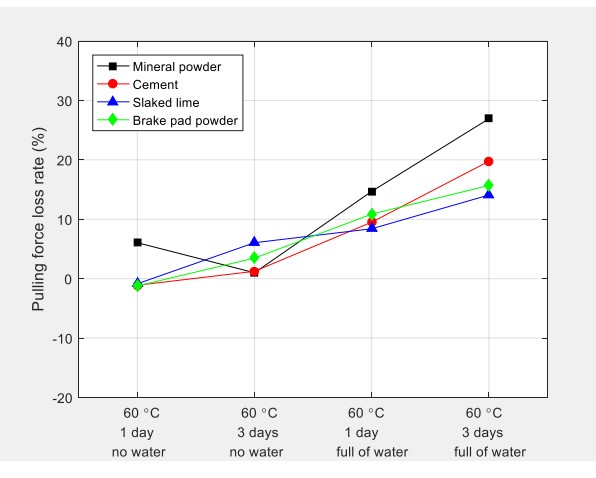

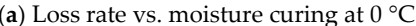

(**a**) Loss rate vs. moisture curing at 0 °C

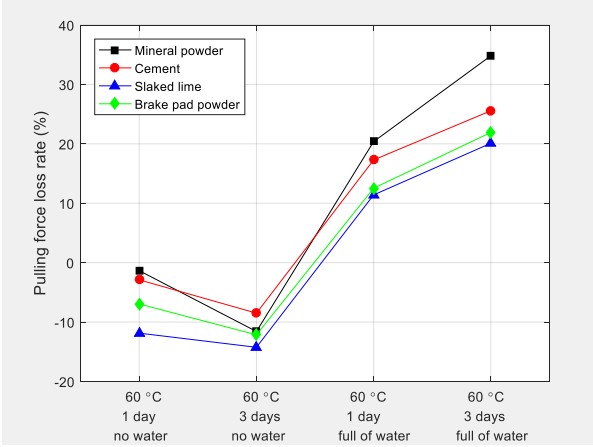

(**b**) Loss rate vs. moisture curing at 60 °C

**Figure 10.** Force reduction rate as a function of moisture curing ($H_2O$ saturation) time at 0 °C and 60 °C.

In general, the graphical results in Figures 9 and 10 indicated a declining response trend with an increase in the moisture ($H_2O$ saturation) curing period for the pulling force and vice versa for the pulling force loss rate. For all the filler types, higher forces

in magnitude and the least loss rates were conversely registered at 1 day curing period without water. As theoretically expected, the degree of force reduction and increase in the loss rate (i.e., damage) were also more distinctive at 60 °C than 0 °C curing temperature. Looking at Figures 9b and 10b at 60 °C for 1 day with full $H_2O$ saturation, the pulling force exhibited a declining response trend with loss rates of about 20.4%, 17.3%, 11.6%, and 12.4% for limestone mineral powder, cement, slaked (whitewashing) lime, and brake pad powder, respectively. After 3 days of $H_2O$ saturation at 60 °C, the force loss rates reached 34.9%, 25.5%, 20.1%, and 22.8%, respectively, with slaked lime registering the highest pulling force and lowest loss rate, respectively. That is the damaging impact of water was higher under the longer moisture conditioning period of 3 days with the $H_2O$ at a higher temperature of 60 °C. Likewise, Figures 9a and 10a exhibited similar graphical response trends to Figures 9b and 10b, with slaked lime generally being superior to the other fillers. After 1 day of $H_2O$ saturation at 0 °C, the force loss rates for the limestone mineral powder, cement, slaked lime, and brake pad powder were 14.7%, 9.3%, 8.4%, and 10.7%, respectively, whilst it was 26.9%, 19.6%, 13.9%, and 15.5% after three days of $H_2O$ saturation at 0 °C.

Overall, these results indicate that after high 60 °C temperature $H_2O$ saturation, the replacement of the limestone mineral powder with the same volume of cement, slaked (hydrated) lime, and brake pad powder can potentially enhance the moisture tolerance and increase the pulling force at the asphalt mortar-aggregate interface, with a corresponding decay in the force loss rate. For the low 0 °C temperature $H_2O$ saturation conditions, the moisture sensitivity and water damage resistance of the asphalt mortar with cement were nearly like that of the slaked (hydrated) lime asphalt mortar. With an increase in the $H_2O$ saturation time from one to three days, however, the asphalt mortar with slaked lime showed better anti-spalling performance. This is because with cement, the hydration reaction occurs in the early stages of the $H_2O$ saturation process. These generated hydration products can potentially lead to forming a new asphalt mortar structure that partially contributes to increasing the adhesive bonding in the shorter term, but undesirably dissipates with more $H_2O$ saturation time. For the two curing temperatures evaluated (i.e., 0 and 60 °C), the pull-out tensile force was highest at 60 °C and increased by more than 8% after 60 °C curing ($H_2O$ saturation) for three days. Compared to the non-saturated curing condition at room temperature, the pulling force at 0 °C without and full water for 1 day in Figure 9a,b had little fluctuations. However, after 0 °C curing in water for three days, the pulling tensile force decreased by about 26%.

From Figure 9, it was generally observed that the pulling force was higher in magnitude under dry (no water) high-temperature conditions and vice versa for the force loss rate in Figure 10. These observations suggest that dry high-temperature conditions are conducive for enhancing the adhesion of the asphalt mortar-aggregate interface, which could be partly attributed to the asphalt's viscoelasticity. This phenomenon was attributed to the intense activity of the asphalt molecules under the dry high-temperature conditions that caused the viscoelastic asphalt to shift towards the viscous phase and readily flow due to the temperature softening effects [44–46]. This probably allowed for better infiltration of the asphalt onto the aggregate pores/surfaces, enabled full contact with the aggregate, and adequate adhesion to the aggregate surfaces.

Under full $H_2O$ saturation conditions, however, the pulling force exhibited a general decreasing response trend as the moisture curing time was increased from one to 3 days, particularly at 60 °C. As theoretically expected, an opposite increasing response trend was observed for the force loss rate, with the 60 °C $H_2O$ saturated specimens enduring quantitatively more decay and loss rate (i.e., all exceeding 10%) than the 0 °C $H_2O$ saturated specimens. This was partly because under $H_2O$ saturated conditions, the tension associated with the water molecules is greater than in the asphalt. Therefore, the polarity of the aggregate will enhance the binding force between the water molecules and aggregate, with water ultimately playing a tearing role between asphalt and aggregates interactions [47]. This phenomenon is further exacerbated if the water is hot at high temperatures such as 60 °C. With more hot $H_2O$ intrusion, the asphalt film potentially de-bonds and falls (i.e.,

strips) off from the aggregate surface, ultimately reducing the interfacial adhesion between the asphalt mortar and aggregate surface. Therefore, the longer the $H_2O$ saturation time and the higher the saturation temperature, the greater the decay in the interfacial adhesion, which is particularly exacerbated under high-temperature wet conditions.

### 3.2. Spalling Rate Test Results and Analysis

### 3.2.1. Structured Light 3-D Scanning of the Interface and Peeling Characterization

The Geomagic Wrap software [34] was used for 3-D modeling. During modeling, the peeling interfaces at parallel distances/spacings of 0.2, 0.4, and 0.6 mm were quantitatively determined, respectively. Thereafter, the peeling rates of the different interfaces were correspondingly computed using Equation (2) for quantifying the moisture sensitivity and water damage resistance of the asphalt-aggregate interface.

Looking at Figure 11, it can be observed that the variational response trends of the spalling sections with different spacings is basically the same, with the spalling rate decreasing with an increase in the spacing. For the different curing ($H_2O$ saturation) conditions that were evaluated, the spalling rate showed an increasing tendency, particularly with the prolongation of the water saturation time to 3 days. This is partly because as the diffusion rate of the water molecules increased with the extension of the water immersion time, the interface phase between the asphalt and aggregate began to become weakened by the invading water molecules [48–50]. The net result is that the asphalt on the aggregate surface was replaced by the water molecules, and hence, a decay in the asphalt-aggregate adhesion. From Figure 11, it is also apparent that this phenomenon, as theoretically expected, was further exacerbated by the high 60 °C saturation/curing temperature with the invading water molecules inherently being more energized.

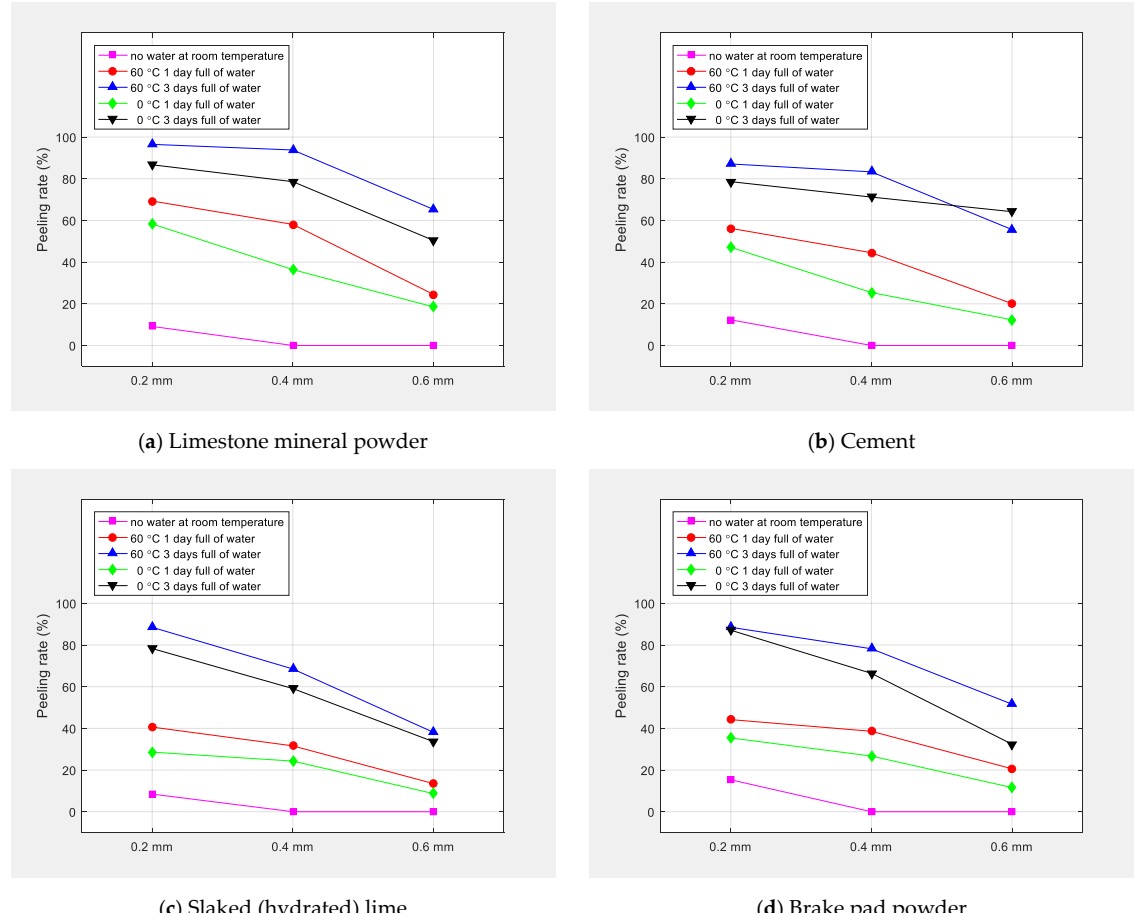

(**a**) Limestone mineral powder　　　　　　　　　　　(**b**) Cement

(**c**) Slaked (hydrated) lime　　　　　　　　　　　(**d**) Brake pad powder

**Figure 11.** Peeling rate results before and after water saturation using the Geomagic Wrap software.

For the asphalt mortar with different fillers as exemplified by the 0.4 mm spalling section, the peeling rates of the slaked lime and brake pad powder after 60 °C moisture ($H_2O$ saturation) curing for one day were 31.6% and 38.6%, respectively. This was significantly lower than the peeling rates associated with the mineral powder and cement fillers. After three days of 60 °C moisture ($H_2O$ saturation) curing, the observed peeling rate relations were as follows: limestone mineral powder (93.8%) > cement (83.3%) > brake pad powder (78.2%) > slaked lime (68.5%). From these results, it is obvious that after 60 °C of $H_2O$ saturation, the spalling rate associated with the cement filler was smaller than that of the limestone mineral powder but greater than the brake pad powder. This suggests that the anti-spalling property of the asphalt mortar can be effectively enhanced by substituting the limestone mineral powder with an equivalent volume of cement.

In Figure 11, the spalling (peeling) rate of the mortar with slaked (hydrated) lime before and after water saturation was the lowest, which indicated that its moisture tolerance and water damage resistance were significantly better than the other three fillers. At one day 0 °C moisture ($H_2O$) saturation condition, the spalling rate (0.4 mm) of the limestone mineral powder, cement, slaked lime, and brake pad powder were 36.4%, 25.4%, 24.3% and 26.7%, respectively. With the extension of the $H_2O$ saturation time to 3 days, however, the spalling rates of the fillers showed noticeable differences—a phenomenal response trend that was consistent with the previous pull-out test results. Overall and as previously inferred, the results in Figure 11 further suggests that the anti-spalling properties of the asphalt mortar with cement and slaked lime fillers are basically the same in the initial stages of low-temperature moisture ($H_2O$ saturation) curing. With time, however, slaked lime exhibited superiority over cement filler.

### 3.2.2. Image-Pro Plus Software Treatment and Spalling Characterization

A digital camera was used for photographically capturing the image of the fracture interface of the pull-out tested specimens. Thereafter, the Image-Pro Plus software [24] was used to determine the spalling rates. The results of these digital imaging and computational analysis are graphed in Figure 12.

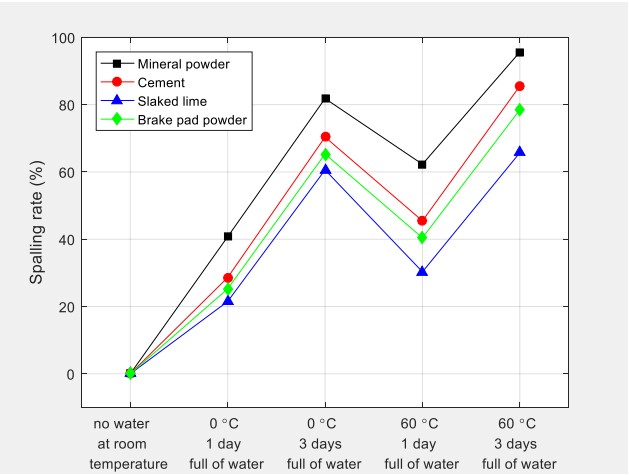

**Figure 12.** Spalling rate results determined using the Image-Pro Plus software.

Looking at Figure 12, it can be observed that the graphical response trends of the spalling (peeling) rate determined using the Image-Pro Plus software yielded similar results to the structured light 3-D scanning results in Figure 11. Under dry (without water) room temperature conditions, the interfaces were all stable with no flaking off and a spalling rate of zero percent (i.e., 0%). After $H_2O$ saturation, however, the interfaces produced different degrees of flaking off and peeling effects, with varying palling rates and almost nearing 100% in the case of the limestone mineral powder after 60 °C moisture conditioning for three days. This was largely attributed to the minimal occurrence of microscopic cohesive failure

of the asphalt mortar at room temperature without water. But with moisture conditioning (i.e., $H_2O$ saturation), however, the moisture has the inherent effects of tearing, displacing, interface weakening, and gap pressurizing between asphalt mortar and aggregate. This makes the asphalt mortar to peel off from the aggregate surface, with the occurrence of adhesive failure.

At elevated temperatures, the effects of water on the asphalt mortar were more obvious. From dry room temperature conditions to $H_2O$ saturation 60 °C for one day, the spalling rates of the limestone mineral powder, cement, slaked lime, and brake pad powder increased to 62.3%, 45.4%, 30.3% and 40.4%, respectively, and continued to rise with the extension of saturation time to three days. For all the curing conditions, the spalling rate generally increased with an increase in the temperature, with more spalling occurring at 60 °C than 0 °C. This can be explained by the fact that at ambient and higher temperature, the asphalt (due to its viscoelastic nature) plays a huge role [51]. When the temperature is about and/or higher than the softening point of the asphalt, the activity of the asphalt molecules is intense, with the asphalt potentially flowing due to temperature softening [48–50]. Likewise, the activities and the movement of the invading water molecules are highly energized at elevated temperatures. Thus, compared with the low temperature curing conditions, the highly energized water molecules are more easily infused within the inside of the asphalt mortar at elevated temperatures. This inherently accelerates and exacerbates the impact of water on the asphalt mortar, with a resultant degradation in the adhesion of the asphalt-aggregate interface [47].

### 3.2.3. Image-Pro Plus Software Treatment and Spalling Characterization

As previously mentioned, parametric corrections and Pearson statistical correlations [38–40] were performed between the spalling rates generated using the Image-Pro Plus software and the structured light 3-D scanning results to verify the accuracy of the interfacial imaging. The results of these parametric analyses are shown in Table 4. Note that although H and S have the same numerical values in the table, their meaning and interpretations are different. H represents the hue (i.e., color or shade) whilst S is the saturation.

**Table 4.** Image-Pro Plus 6.0 parametric analysis and Pearson's correlation coefficients.

| Interface | Item | Tonal H | Saturation S | The Intensity of the I | Correlation Coefficient ($R^2$) |
|---|---|---|---|---|---|
| 0.2 mm | Mineral powder mortar | 0 ~ 255 | 0 ~ 255 | 0 ~ 95 | 0.985 |
| | Cement mortar | 0 ~ 255 | 0 ~ 255 | 0 ~ 110 | 0.990 |
| | Slaked lime mortar | 0 ~ 255 | 0 ~ 255 | 0 ~ 75 | 0.998 |
| | Brake pad mortar | 0 ~ 255 | 0 ~ 255 | 0 ~ 60 | 0.986 |
| 0.4 mm | Mineral powder mortar | 0 ~ 255 | 0 ~ 255 | 0 ~ 120 | 0.995 |
| | Cement mortar | 0 ~ 255 | 0 ~ 255 | 0 ~ 130 | 0.999 |
| | Slaked lime mortar | 0 ~ 255 | 0 ~ 255 | 0 ~ 105 | 0.999 |
| | Brake pad mortar | 0 ~ 255 | 0 ~ 255 | 0 ~ 95 | 0.999 |
| 0.6 mm | Mineral powder mortar | 0 ~ 255 | 0 ~ 255 | 0 ~ 140 | 0.950 |
| | Cement mortar | 0 ~ 255 | 0 ~ 255 | 0 ~ 165 | 0.960 |
| | Slaked lime mortar | 0 ~ 255 | 0 ~ 255 | 0 ~ 130 | 0.998 |
| | Brake pad mortar | 0 ~ 255 | 0 ~ 255 | 0 ~ 110 | 0.972 |

At $R^2$ = 99.9%, it can be seen in Table 4 that the correlation coefficients for the 0.4 mm interface were the highest, which is more consistent with the actual peeling surfaces. Therefore, the parameters of the Image-Pro Plus 6.0 software [23] were corrected and modified using the 0.4 mm peeling interface as the reference datum. This aided to quantitatively express the moisture sensitivity and water damage resistance of the asphalt mortar with different fillers more accurately. The results of these correlation analyses and the corrected spalling rates are shown in Figure 13.

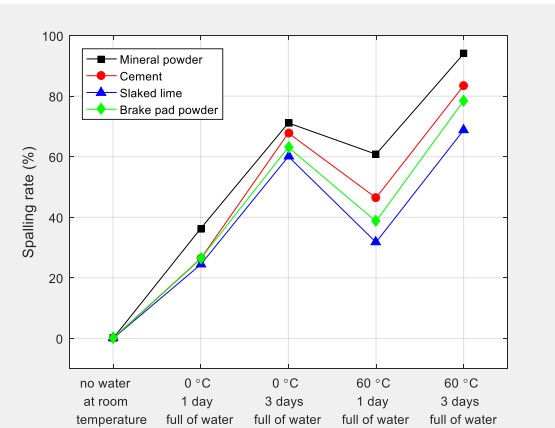

**Figure 13.** Corrected spalling rates.

The results in Table 4 indicated that for the peeling (spalling) rate analysis, a digital camera can be directly used to photographically scan and collect the images on the fracture surfaces of the pull-out tested specimens. The digital images can then be processed using the Image-Pro Plus software for parametric corrections and Pearson statistical correlations. In this way, the spalling rate determination can be more rapid, reliable, and accurate. Overall, the corrected spalling rates in Figure 13 showed a similar response trend as was previously observed; with the adhesion degrading over time (particularly at high temperature $H_2O$-curing conditions) and the slaked lime exhibiting superiority over the other filler materials.

### 3.3. Scanning Electron Microscope (SEM) Results and Analysis

For easy visual comparison purposes and to increase the identification accuracy of the microscopic analysis, only specimens at normal temperature (i.e., control conditions) and 60 °C moisture curing (i.e., worst conditions) were selected for SEM imaging. The corresponding microscopic imaging results of the failure interfaces are shown in Figure 14.

Under dry conditions, the limestone mineral powder get immersed within the asphalt, forming a layer of dense asphalt membrane or film with a high cohesive bonding force that is visually exemplified in Figure 14a. After full water saturation at 60 °C, as the moisture diffuses and infiltrates through the asphalt membrane, part of the limestone mineral powder particles shown in Figure 14b get exposed on the outside of asphalt and consequently weakens the interfacial bond. And this partially contributes to the ultimate decay and degradation in the interfacial adhesion.

When the asphalt mortar with cement filler is not saturated, the cement particles appear massive as exemplified in Figure 14c. After the hydration reaction in the presence of water, some cement particles form needle-like crystals. The resulting crystal hydrate and cement particles form a 3-D structure exemplified in Figure 14d. This is largely due to the formation of calcium silicate hydrate, calcium hydroxide, and calcium sulfanilamide hydrate after the hydration reaction of part of the cement [52]. These hydration products along with the asphalt mortar and by-products get interwoven with each other to form a new asphalt mortar structure albeit of inferior adhesion and interfacial bonding.

As shown in Figure 14e, when the asphalt mortar mixed with slaked (hydrated) lime is not $H_2O$ saturated, the lime particles get wrapped by asphalt with laterally no visible interfacial zone. However, after the slaked lime is saturated with water, the products of the chemical reaction between the lime and asphalt along with the unreacted slaked lime particles form more stable block structures exemplified in Figure 14f. This is because calcium hydroxide, the main component in the slaked lime, gets into contact with the carboxylic acid and SARA (namely Saturate, Aromatic, Resin, and Asphaltene) in the asphalt [53]. This fundamentally leads to chemical reactions that produce products with strong absorbability with the potential to firmly adhere to the aggregate surface without

peeling off, thus increasing the moisture tolerance, water damage resistance, and anti-spalling properties of the asphalt mortar.

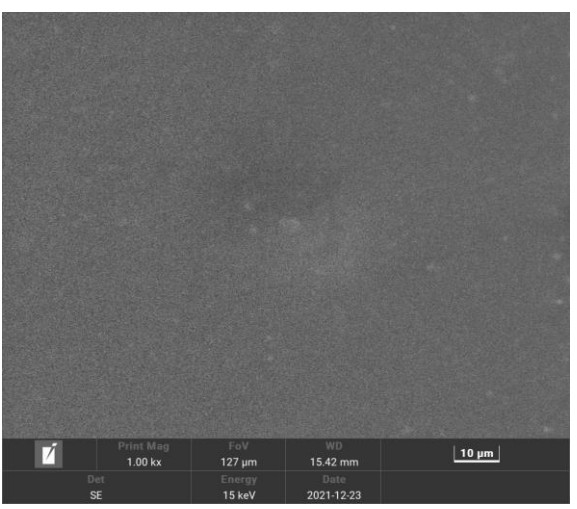

(**a**) Limestone at room temperature with no water

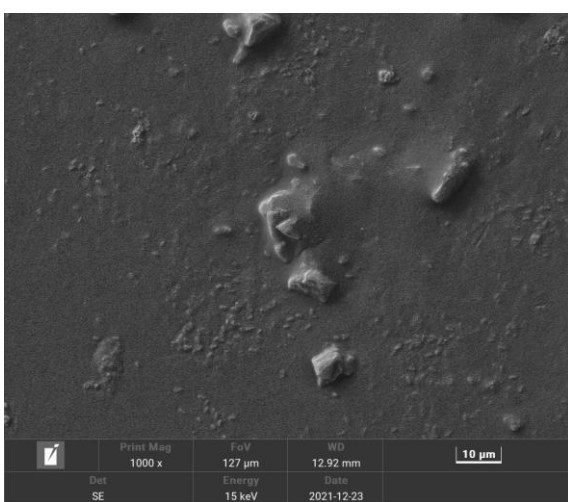

(**b**) Limestone saturated $H_2O$ at 60 °C for 3 days

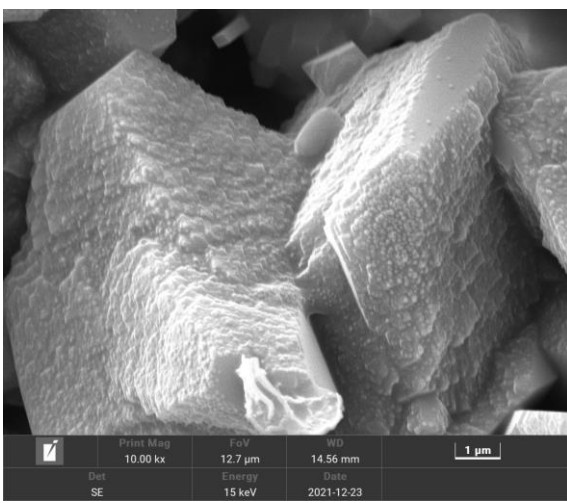

(**c**) Cement at room temperature with no water

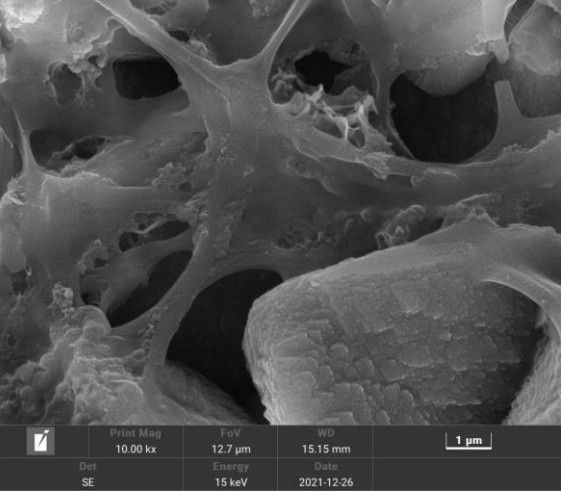

(**d**) Cement saturated with $H_2O$ at 60 °C for 3 days

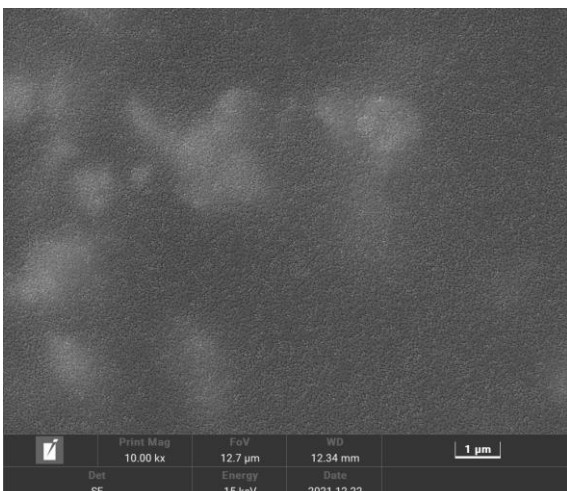

(**e**) Lime at room temperature with no water

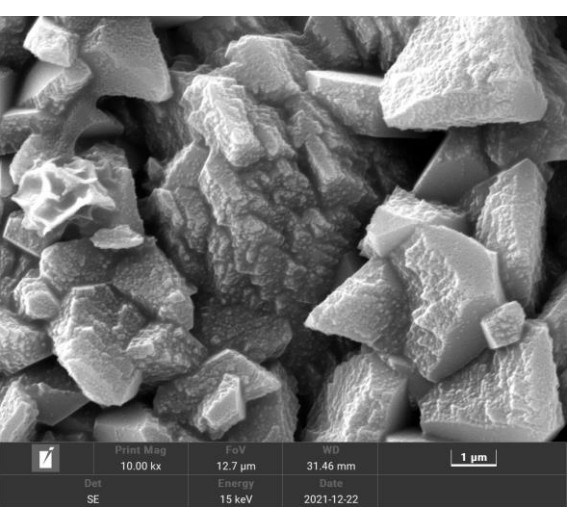

(**f**) Lime saturated $H_2O$ at 60 °C for 3 days

**Figure 14.** *Cont.*

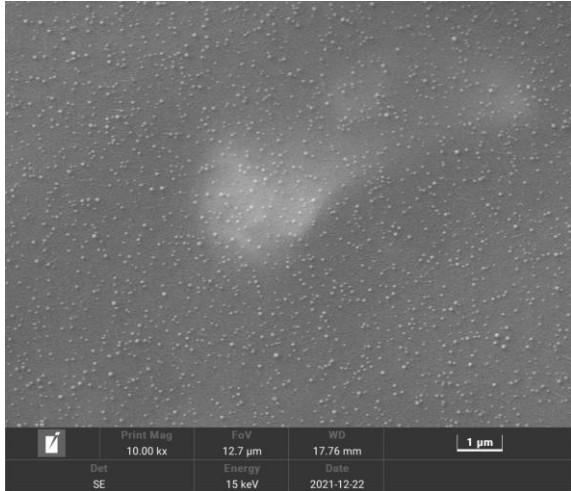
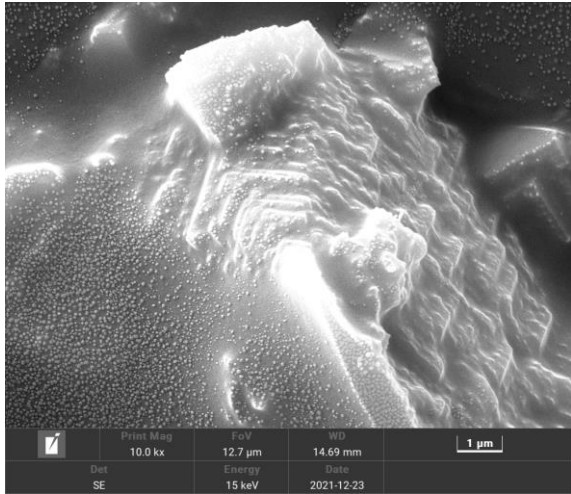

(**g**) Brake pad at room temperature with no water       (**h**) Brake pad saturated H₂O at 60 °C for 3 days

**Figure 14.** SEM microscopic morphology of asphalt mortar before and after water saturation.

Compared to other filler materials, the thickness of the asphalt film becomes thinner when the brake pad powder is saturated with water—see Figure 14g,h. Concurrently, because it contains more calcium oxide, it induces a hydration reaction with water and generates stronger adsorbable products that to some degree enhances the anti-spalling performance of the asphalt mortar [30]. Because cement, slaked lime, and brake pad powder have stronger chemical reactions in water, their corresponding asphalt mortar water stability is better than that of the mineral powder (limestone) asphalt mortar. Furthermore, the chemical reaction products of slaked lime, brake pad powder, and water interacts better with the asphalt, and so, the water stability of the asphalt mortar is better than that of the cement asphalt mortar. Compared to slaked (hydrated) lime, brake pad powder has more impurities with relatively lower pure alkali-containing substances. So, the resultant water stability of its asphalt mortar is correspondingly lower than that of the slaked lime asphalt mortar. Overall, the brake pad powder was ranked second best performer after slaked (hydrated) lime. The poorest performer at the bottom-most rank was the limestone mineral powder.

## 4. Summary and Conclusions

This study was conducted to comparatively evaluate and quantitatively characterize the moisture sensitivity and water damage resistance of the interfacial bonding between asphalt mortar and aggregate fillers. Using an in-house custom developed water-temperature coupling setup, numerous laboratory pull-out tests were carried out on the asphalt mortar with four different fillers, namely limestone mineral powder, P·O 42.5 cement, slaked (hydrated) lime, and waste brake pad powder, respectively. The variables investigated included moisture wet-curing conditions, temperature, and filler type. The Image-Pro Plus software, 3-D digital imaging, and scanning electron microscope (SEM) were used to quantify the spalling rate and the surface micromorphology of the asphalt mortar-aggregate filler interface before and after water saturation, respectively. The key findings, conclusions, and recommendations drawn from the study are summarized below:

(1) As was evident from the declining pull-out tensile force and the increasing force loss rate, respectively, the moisture tolerance and water damage resistance of the asphalt mortar decreased in the presence of water. The adhesion and interfacial bonding strength generally degraded with time and being pronounced at elevated H₂O-curing temperatures. Thus, it was concluded that the coupling effects of water, age (i.e., saturation time), and temperature are detrimental to the adhesion and bonding strength of the asphalt-aggregate interface.

(2) With respect to filler material comparisons, the results indicated superiority for slaked (hydrated) lime followed by the brake pad powder. The poorest performer with the

highest moisture sensitivity and water damage susceptibility was the limestone mineral powder. On this basis, this study recommends the use of slaked (hydrated) lime and brake pad powder in high-temperature rainy-wet environments. Cement (along with limestone mineral powder_, on the other hand, would be the tentatively suggested option for low-temperature low-rain environments.

(3) The 3-D image analysis, Pearson statistical correlations, and SEM imaging all collectively indicated that the coupling effects of water, saturation time, and high $H_2O$-curing temperature detrimentally degraded the interfacial bonding of the asphalt mortar, with a corresponding increase in the peeling off rate. However, the moisture tolerance and $H_2O$ damage resistance were observed to significantly improve under dry low-temperature conditions with minimal water saturation periods. Furthermore, it was also found that the 0.4 mm interface was the appropriate reference datum for quantitative analysis of the interfacial fracture images and Pearson statistical correlation.

Overall, this study valuably contributes to the state-of-the-art literature enrichment through provision of a supplementary datum for quantitatively characterization the asphalt-aggregate interfacial bonding as a function of $H_2O$ and temperature using the pull-out test, 3-D digital photo-graphics, and SEM imaging. The results plausibly indicated superior laboratory performance for slaked (hydrated) lime; consecutively followed by brake pad powder, cement, and limestone mineral powder (poorest performer), respectively. For future follow-up studies, more laboratory variables such as different base asphalts (in lieu of Pen #70), material types, filler type/contents, $H_2O$ curing conditions, high test temperatures, test methods (such as shear), etc., should be included in the study matrix along with field validation to further complement and substantiate the findings reported herein.

**Author Contributions:** Conceptualization, F.X.; Methodology, W.G.; Software, X.N.; Validation, W.G.; Formal analysis, P.X.; Investigation, F.X.; Data curation, H.E.; Writing—original draft, H.C.; Writing—review & editing, H.C.; Coordinate all parties, R.G. and Y.Z. All authors have read and agreed to the published version of the manuscript.

**Funding:** This research received no external funding.

**Institutional Review Board Statement:** Not applicable.

**Informed Consent Statement:** Not applicable.

**Data Availability Statement:** The main authors have shared their research data.

**Conflicts of Interest:** The authors declare no conflict of interest.

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
