# Peer review of "Moisture Sensitivity Evaluation of the Asphalt Mortar-Aggregate Filler Interface Using Pull-Out Testing and 3-D Structural Imaging"

_coatings, doi:10.3390/coatings13050868_

Round 1
Reviewer 1 Report
Line 112: the author has used Chinese standards for the characterization of asphalt binder, there is a need to prepare a comparison of the mentioned standards with the ASTM or AASHTO standards as these are used mostly worldwide.
Table 2 to 5 need to be improved; it is suggested to prepare a single table for all the fillers so the results of XRF analysis can be easily compared. Also, table 6 should be included in the same table or improve this.
Line 132: there is an error written in the text. The author needs to improve the text and formatting.
Line 135: the preparation of asphalt slurry is a standard method, so authors needs to mention which standard has been followed to prepare the slurry and from where the percentage, temperature, and rotation speed for mixing is determined. Is that from a previous study or a standard?
Line 143: What is the reason to keep the control sample at 60°C in oven for up to 3 days? Also, for the other samples placed in water bath at 60°C what is the time involved?
Line 146: the author has mentioned a control group of samples. Why has the author not performed any tests on pure binder without the addition of any filler?
Line 152: the author again fails to mention any testing standard or process that has been followed to conduct the test. Is the jig made specifically for this experiment or it is from the certain configuration of UTM instrumentation, please mention the details of the testing procedure.
Improve the quality of Figure 1, the assemble can not be seen clearly in the pictures.
Line 169 and 172 the reference is missing, please update.
Improve Figure 4 as this is not properly showing the image processing using Image-Pro plus 6.0 Software
Line 222: the author is suggested to convert the paragraph into bullet points to better understand the steps. Also, try to include a flow chart for the complete process, which makes it easier to understand and a much clear representation of the steps followed.
There are many errors in reference sources throughout the text so major revision is required for the paper.
Figure 5 needs to be improved, the text is not clear and understandable in the flow diagram.
The selection of test temperatures needs to be justified as the purpose is to check moisture sensitivity and 0°C is not a suitable temperature for that.
As the temperature increases the visco-elastic properties of the binder start to show. So, the selection of temperature is very important, the author needs to justify the selection of test temperatures.
Table 7 shows the same values for 3rd and 4th columns. Why have they been added to the table?
Figure 12: the data can better be represented by a bar graph to show the comparison rather than using a line graph. What is the purpose of using a line graph in this context?
The result are not conclusive, please improve them.

Reviewer 2 Report
The paper Moisture Sensitivity Evaluation of the Asphalt Mortar-Aggregate Filler Interface Using Pull-Out Testing and 3-D Structural Imaging is suitable for publication, and well presented and written by the authors, I recommend its publication, after minor corrections:
(a) The state of the art is still limited, the authors must explore and deepen the theme approach;
(b) The discussion of the results is still limited, the authors need to go deeper with comparisons with other studies in the international literature in the area;
(c) The conclusion must be reformulated in light of the objectives proposed by the authors.
Reviewer 3 Report
The paper "Moisture Sensitivity Evaluation of the Asphalt Mortar-Aggregate Filler Interface Using Pull-Out Testing and 3-D Structural Imaging" presents a relevant theme and within the scope of this journal, and can be considered after some corrections suggested below:
(a) The abstract is generally well written, however in terms of content it is generic, i.e., the authors lack an in-depth study of the quantitative results of this research;
(b) Scientific innovation is limited in the introduction of the paper, the authors must go deeper and detail what this research differs from countless others that exist on this topic, this must be evidenced together with the objectives at the end of the introduction;
(c) The state of the art of the evaluated topic needs to be improved by the authors, note that some topics are absent and need to be known with current research;
(d) Add an experimental flowchart to the methodology step of this paper;
(e) “From Figure 8, it was generally observed that the pulling force was higher in magni- 343 tude under dry condition with no water and high temperature conditions – and vice versa 344 for the force loss rate in Figure 9. These observations suggests that dry high-temperature 345 conditions are conducive for enhancing the adhesion of the asphalt mortar-aggregate in- 346 terface. Under full H2O saturation conditions, the pulling force exhibited a decreasing 347 response trend with more accelerated decay occurring at elevated temperatures as the 348 moisture curing time was increased to 3 days. As theoretically, an opposite increasing 349 response trend was observed for the force loss rate.” Explain this part of the text better.
Reviewer 4 Report
1-please improve the literature review by addition of recent published papers
2-improve the results section by addition of more discusion
3-improve the discussion by link the results of SEM by pull off results
4- improve the conclusion part
Round 2
Reviewer 1 Report
Thanks.
Reviewer 3 Report
The authors have made all the indicated corrections.